# Assessment of Life Cycle Modeling Systems as Prediction Tools for a Possible Attenuation of Recombinant Ebola Viruses

**DOI:** 10.3390/v14051044

**Published:** 2022-05-13

**Authors:** Bianca S. Bodmer, Thomas Hoenen

**Affiliations:** Institute for Molecular Virology and Cell Biology, Friedrich-Loeffler-Institut, 17493 Greifswald, Germany; bianca.bodmer@fli.de

**Keywords:** Ebola virus, filovirus, reverse genetics, life cycle modeling system, minigenome system, trVLP system, virus rescue, tagged virus, reporter virus, green fluorescent protein

## Abstract

Ebola virus (EBOV) causes hemorrhagic fever in humans with high case fatality rates. In the past, a number of recombinant EBOVs expressing different reporters from additional transcription units or as fusion proteins have been rescued. These viruses are important tools for the study of EBOV, and their uses include high throughput screening approaches, the analysis of intercellular localization of viral proteins and of tissue distribution of viruses, and the study of pathogenesis in vivo. However, they all show, at least in vivo, attenuation compared to wild type virus, and the basis of this attenuation is only poorly understood. Unfortunately, rescue of these viruses is a lengthy and not always successful process, and working with them is restricted to biosafety level (BSL)-4 laboratories, so that the search for non-attenuated reporter-expressing EBOVs remains challenging. However, several life cycle modeling systems have been developed to mimic different aspects of the filovirus life cycle under BSL-1 or -2 conditions, but it remains unclear whether these systems can be used to predict the viability and possible attenuation of recombinant EBOVs. To address this question, we systematically fused N- or C-terminally either a flag-HA tag or a green fluorescent protein (GFP) to different EBOV proteins, and analyzed the impact of these additions with respect to protein function in life cycle modeling systems. Based on these results, selected recombinant EBOVs encoding these tags/proteins were then rescued and characterized for a possible attenuation in vitro, and results compared with data from the life cycle modeling systems. While the results for the small molecular tags showed mostly good concordance, GFP-expressing viruses were more attenuated than expected based on the results from the life cycle modeling system, demonstrating a limitation of these systems and emphasizing the importance of work with infectious virus. Nevertheless, life cycle modeling system remain useful tools to exclude non-viable tagging strategies.

## 1. Introduction

Ebola virus (EBOV) belongs to the family of filoviruses that currently comprises four genera found in mammals [1,2]. Two of these genera, the ebolaviruses and the marburgviruses, include viruses that cause severe hemorrhagic fever in humans, such as EBOV and Marburg virus (MARV), with case fatality rates of up to 90% [3]. The most virulent ebolavirus is EBOV (species *Zaire ebolavirus*), and this virus is responsible for the two most extensive outbreaks of filovirus disease, which occurred in 2014–2016 and 2018–2020.

The negative-sense single-stranded RNA genome of EBOV encodes seven structural proteins. The nucleoprotein (NP) encapsidates the viral genome [4], and forms together with the genome as well as the transcriptional activator viral protein (VP)30, the polymerase co-factor VP35, and the polymerase (L) the ribonucleoprotein complex (RNP). These four RNP proteins are necessary and sufficient to facilitate viral transcription and replication [5]. VP24 is an interferon antagonist and necessary for condensation of the RNP complex [6,7]. The matrix protein VP40 orchestrates morphology and budding of viral particles at the plasma membrane of infected cells, and the glycoprotein (GP) mediates attachment and fusion of virus particles with target cells [8,9].

While work with virus isolates or recombinant viruses produced by full-length clone systems is restricted to biosafety level (BSL)-4 conditions, reverse genetics-based life cycle modeling systems can mimic specific aspects of the EBOV life cycle under BSL-1 or -2 conditions (reviewed in [10]). The simplest of these life cycle modeling systems is the monocistronic (1-cis) minigenome system, which models viral RNA synthesis and protein expression. A 1-cis minigenome contains the noncoding leader and trailer regions found at the ends of the EBOV genome; however, all viral genes are removed and replaced by a reporter gene, encoding, for example, a fluorescent protein or a luciferase. Transfection of target cells with a plasmid encoding for this 1-cis minigenome in combination with expression plasmids for the 4 RNP proteins results in replication and transcription of the minigenome by the RNP proteins and subsequent expression of the reporter protein. In a tetracistronic (4-cis) minigenome system, which is based on a 1-cis minigenome system, the viral genes for VP24, VP40, and GP are added to the minigenome, so that all viral proteins are now expressed, and transcription and replication-competent virus-like particles (trVLPs) are formed. These trVLPs incorporate the minigenome in form of RNPs and can be used to infect target cells that have been pre-transfected to express the RNP proteins, so that replication and transcription of the minigenome can take place also in these target cells. Reporter activity in both systems reflects the efficiency of replication and transcription of the minigenome, and, in case of the 4-cis minigenome system, reporter activity in target cells also reflects morphogenesis, budding, and entry.

The reverse genetics-based full-length clone system allows the generation of recombinant EBOV for in vitro and in vivo studies of the whole virus life cycle, which can be aided by the use of molecular tags or reporter genes. Recombinant viruses with small molecular tags could, for example, be used for identification of host factors interacting with viral proteins via co-immunoprecipitation followed by mass spectrometry, while applications for recombinant viruses encoding luciferases or fluorescence proteins range from high throughput screenings for antivirals [11,12] or interacting host-factors [13] to analysis of cellular processes via life cell imaging in vitro [14,15,16], as well as virus spread to organs or specific cell types for pathogenicity studies in vivo [17,18].

However, rescue and characterization of such recombinant viruses is a lengthy process under high containment conditions, and addition of a reporter gene, depending on the insertion strategy, might not lead to a rescuable virus, or result in a virus that is attenuated. Therefore, we tested whether life cycle modeling systems can predict the viability or attenuation of recombinant viruses with tagged viral proteins. Initially, these experiments were done with a flag-HA tag, and rescue of all five tested recombinant viruses predicted to be viable by life cycle modeling systems was possible, including a non-attenuated rgEBOV-L-flag-HA. Furthermore, fusion of an enhanced green fluorescent protein (GFP), as a model for fluorescence proteins, to the viral protein sites that tolerated tagging with the smaller flag-HA tag either directly or via a 2A peptide from *Thosea asigna* virus (T2A) also produced three replicating recombinant viruses; however, here, attenuation was more severe than expected based on the minigenome data.

## 2. Materials and Methods

### 2.1. Cells

293T (human embryonic kidney cells, Collection of Cell Lines in Veterinary Medicine CCLV-RIE1018), Huh7 (human hepatocarcinoma cells, kindly provided by Stephan Becker, Philipps University Marburg), and Vero E6 (African green monkey kidney cells, kindly provided by Stephan Becker, Philipps University Marburg) cells were cultured at 37 °C and 5% CO_2_ in Dulbecco’s modified Eagle’s medium (DMEM; ThermoFisher Scientific, Darmstadt, Germany) supplemented with 10% fetal calf serum (FCS), 100 U/mL penicillin, 100 µg/mL streptomycin (PS; ThermoFisher Scientific), and 1× GlutaMAX (ThermoFisher Scientific).

### 2.2. Plasmids

Components for the 1-cis and 4-cis minigenome assay, including expression plasmids encoding the EBOV RNP proteins, T7 polymerase, firefly luciferase, T-cell immunoglobulin and mucin domain 1 (TIM-1), and the EBOV minigenomes (p1cis-EBOV-vRNA-hrLuc, p4cis-EBOV-vRNA-hrLuc) have been previously described [7,19].

N-terminally flag-HA-tagged RNP constructs were cloned as described elsewhere [20]. For cloning of C-terminally tagged RNP proteins, a PCR and TypeIIS-based cloning approach was used for insertion into a pCAGGS-C-term-flag-HA-iBsmBI plasmid, which included a *BsmB*I insertion site in front of a flag-HA-tag (DYKDDDDKLDGGYPYDVPDYA). Cloning of the C-terminally (T2A-)GFP-tagged RNP constructs was performed by inserting RNP protein genes into pCAGGS-T2A-oligo-GFP or pCAGGS-oligo-GFP, which encode a GSG-linker in front of the GFP, using standard cloning methods. N-terminally GFP-tagged plasmids were cloned via hot-fusion technology [21]. Flag-HA- or (T2A-) GFP-tagged RNP proteins for full-length EBOV genome plasmids were first inserted in cassette vectors [19], which were then exchanged in the full-length EBOV genome plasmid pAmp-rgZ-dSXBS via hot-fusion technology and standard cloning methods. Detailed cloning strategies are available on request.

### 2.3. Minigenome and trVLP Assays

Minigenome assays were performed in 12-well format as previously described [7,19]. Briefly, 293T cells were seeded for approximately 50% confluency and transfected using 3 μL per μg DNA of TransIT LT-1 (Mirus Bio LLC, Madison, WI, USA) with pCAGGS expression plasmids encoding for NP (62.5 ng/well), VP35 (62.5 ng/well), VP30 (37.5 ng/well), and L (500 ng/well), or the respective flag-HA- or (T2A-)GFP-tagged expression plasmids for RNP proteins, as well as expression plasmids for T7-polymerase (125 ng/well) and a 1-cis minigenome plasmid (125 ng/well) or 4-cis minigenome plasmid (125 ng/well) following the manufacturer’s instructions. As a negative control, the expression plasmid for L was omitted and replaced by pCAGGS-eGFP. Twenty-four hours after transfection, medium was exchanged with 1 mL (1-cis minigenome) or 2 mL (4-cis minigenome) DMEM supplemented with 5% fetal calf serum (FCS), 100 U/mL penicillin, 100 µg/mL streptomycin, and 1× GlutaMAX. Cells were lysed 48 h (1-cis minigenome assays) or 72 h (4-cis minigenome assay) after transfection in 200 µL 1× Lysis Juice (PJK) for 10 min. Cell debris was removed by centrifugation for 3 min at 10,000× *g*. Then, 40 μL lysate was added to 40 μL Beetle Juice (PJK) or 40 μL Renilla Glo Juice (PJK) in opaque 96-well plates, and luminescence was measured using a Glomax Multi (Promega) microplate reader. Renilla luciferase activities were normalized to Firefly luciferase activities. In the 4-cis minigenome assay, p1 cells were additionally pre-transfected with RNP expression plasmids and an expression plasmid for TIM-1 (125 ng/well) as described above 24 h prior to harvesting of the p0 cells, and 24 h later they were infected with 1.5 mL clarified (5 min, 800× *g*) supernatants from p0 cells. p1 cells were lysed 72 h post infection as described above for p0 cells.

### 2.4. Fluorescence Microscopy

Fluorescence images of transfected cells were taken 48 h post transfection of 4-cis p0 cells using a Thunder Imaging system (Leica). Virus-infected cells were imaged daily with a Vert.A1 microscope (Zeiss) and an EOS 1000D camera (Canon).

### 2.5. Viruses Rescue and Stock Production

Full-length cloning plasmids were based on Zaire ebolavirus rec/COD/1976/Mayinga-rgEBOV (GenBank accession number KF827427.1, rgEBOV) [22]. Rescue of the flag-HA- or (T2A-) GFP-tagged viruses was done as previously described [23]. Briefly, Huh7 cells were seeded in 6-wells for about 50% confluency the next day. Cells were transfected with 3 μL per μg DNA of TransIT LT-1 using pCAGGS-based expression plasmids for NP (125 ng/well), VP35 (125 ng/well), VP30 (75 ng/well), L (1000 ng/well), T7-polymerase (250 ng/well), and respective full-length genome plasmids (250 ng/well). Per rescue attempt, six wells were transfected, but in one well the expression plasmid for L was omitted as a negative control. Twenty-four hours post transfection, the medium was exchanged to 4 mL/well DMEM supplemented with 5% fetal calf serum (FCS), 100 U/mL penicillin, 100 µg/mL streptomycin, and 1× GlutaMAX. One week after transfection, 500 µL supernatants from transfected Huh7 were passaged onto VeroE6 cells (about 90% confluency in 6-well plates). Cells were checked for cytopathic effect or green fluorescence about 2 weeks after passaging. Virus clones were then propagated in VeroE6 cells, and virus titers were determined by 50% tissue culture infectious dose (TCID_50_) assay. The sequence was confirmed by Sanger sequencing as described elsewhere [23]. All work with infectious virus was performed under BSL-4 conditions at the Friedrich-Loeffler-Institut (Federal Research Institute of Animal Health, Greifswald Insel-Riems, Greifswald, Germany) following approved standard operating procedures.

### 2.6. Growth Kinetics of Recombinant Viruses and Titration

Vero E6 cells (about 90% confluence) were infected with 1 × 10^4^ TCID_50_ EBOV (MOI = 0.01) on day 0 for 1 h, as previously described [24]. Subsequently, cells were washed three times with serum-free medium, and then cultured in 4 mL DMEM supplemented with 5% fetal calf serum (FCS), 100 U/mL penicillin, 100 µg/mL streptomycin, and 1× GlutaMAX. From day 0 to 6 post-infection, 500 µL of supernatant was harvested daily, stored at −80 °C until titration, and replaced with 500 µL fresh culture medium. Virus titers in supernatants from days 0 to 6 post-infection were determined by TCID_50_ assay in Vero E6 cells [25]. The lower limit of detection was 6.3 × 10^1^ TCID_50_/_mL_.

### 2.7. Statistical Analysis

One-way ANOVA (for minigenome and trVLP assays) or two-way ANOVA (for growth kinetics) with Dunnett’s multiple comparison tests were performed using GraphPad Prism 8.1.0.

## 3. Results

### 3.1. N- and C-Terminally Flag-HA-Tagging of EBOV Proteins Has Different Effects on Their Function

The use of life cycle modeling systems to assess the functionality of tagged EBOV proteins before attempting to rescue the respective recombinant rgEBOVs could prevent lengthy cloning and subsequent unsuccessful rescue attempts under high containment conditions. To assess the viability of this strategy, we tested a number of tagged proteins in life cycle modeling systems, before cloning corresponding full-length genome plasmids and rescuing the respective viruses.

As a first tagging strategy, a flag-HA-tag was fused to either the N- or C-terminus of the EBOV RNP proteins, as a model for small molecular tags. These were then tested for functionality in 1-cis and 4-cis minigenome assays (Figure 1) to assess impairment of genome replication and transcription, as well as effects on packaging, budding, and entry processes. Directly tagging the N-terminus of the EBOV RNP proteins VP30 and L, but not of NP and VP35, resulted in reduced reporter activity in the 1-cis minigenome assay (Figure 1a) or the 4-cis minigenome assay (Figure 1b) in p0 cells.

In p1 cells (Figure 1c), reporter activity significantly dropped for all N-terminally tagged constructs. Here, reduction in reporter activity of tagged NP and VP30 were more pronounced, with VP30 being only 2.6-fold over background (-L) levels. In contrast, fusion of a flag-HA tag to the C-terminus of any of the EBOV RNP proteins did not reduce reporter activity in the 1-cis minigenome system (Figure 1d) or 4-cis minigenome system in p0 cells (Figure 1e). However, in p1 cells of the 4-cis minigenome system (Figure 1f), reporter activity in presence of VP35-flag-HA was dramatically reduced, with a 40-fold drop compared to wt activity, and also reporter activity in case of NP-flag-HA showed some reduction.

### 3.2. Direct Fusion of EBOV Proteins with a Flag-HA-Tag Results in Rescueable rgEBOV with Different Levels of Attenuation

Based on the results from the life cycle modeling systems, for cloning and rescue of infectious virus, we chose five variants. For each of the RNP proteins, the tagging-variant which showed less impact in the life cycle modeling systems was selected. In case of NP, we decided to proceed with both variants, as there was no clear difference between them in life cycle modeling systems (Figure 1c,f).

An overview of the genome structure of rescued viruses is shown in Figure 2a. Rescue of all viruses was successful, with a rescue efficiency of 100% (Figure 2a), and we could produce stocks with titers in the range of 10^7^ TCID_50_/_mL_, except for rgEBOV-VP30-flag-HA, which had a lower stock titer of 2 × 10^5^ TCID_50_/_mL_ despite showing decent development of cytopathic effect at time of harvesting (not shown). Sequences of all viruses were confirmed via Sanger sequencing.

To assess a possible attenuation in vitro, we performed growth kinetics (Figure 2b). The tagged viruses fell into three groups: non-attenuated (rgEBOV and rgEBOV-L-flag-HA), slightly attenuated (rgEBOV-VP30-flag-HA and rgEBOV-NP-flag-HA), and highly attenuated (rgEBOV-flag-HA-NP and rgEBOV-flag-HA-VP35) (summarized in Figure 2c). 

In most cases, attenuation corresponded to the minigenome data, and, in particular, rgEBOV-L-flag-HA, which had shown no reduction in reporter activity in life cycle modeling systems, showed also no in vitro attenuation when compared with rgEBOV kinetics. However, despite having no effect in life cycle modeling systems, addition of a flag-HA tag at the C-terminus of VP30 resulted in a slightly attenuated rgEBOV-VP30-flag-HA virus.

### 3.3. Fusion of eGFP to EBOV RNP Proteins Impacts Protein Functionality

Based on the results with the flag-HA tag, we cloned expression plasmids for GFP-tagged RNP proteins for the most promising tagging strategies. In these constructs GFP was either directly fused to the viral protein, or connected by a T2A site, which results in cotranslational ribosomal stalling producing two proteins from a single open reading frame, sometimes also referred to as “self-cleavage”, and which with ~90% “cleavage” efficiency has been shown to be the most efficient among the 2A peptides [26,27]. Since the T2A site leaves the amino acids GSGEGRGSLLTCGDVEENPG at the C-terminus of the upstream protein, only RNP proteins that allowed C-terminal addition of the flag-HA tag were chosen for this strategy.

Expression of these constructs replacing the respective RNP protein in a 1-cis minigenome system showed green fluorescence of differing intensity, and particularly fluorescence intensity of GFP-L was weaker than that of the other fusion proteins, and only visible after adjusting the exposure time (Figure 3a). With respect to protein functionality in a monocistronic minigenome assay, in case of NP direct tagging reduced minigenome reporter activity by about 20-fold regardless of which terminus was used for tagging, while tagging of L reduced reporter activity by approximately five-fold. The other fusion proteins showed no significant reduction in minigenome reporter activity.

In the 4-cis minigenome, p0 data resembled data from the 1-cis minigenome assay, as expected (Figure 3c). However, p1 data (Figure 3d) clearly showed that directly GFP-tagged NP is non-functional, independent of the tagging site, as reporter activity dropped to background levels (-L). N-terminal GFP-tagging of VP35 also strongly reduced protein functionality. In contrast, functionality of L and VP30 was not influenced by C-terminal addition of GFP, independent of whether this was a direct fusion or fusion via a T2A site.

Based on the minigenome results, we attempted to rescue five different GFP-tagged viruses either with or without a T2A-site, as depicted in Figure 4a. Rescue was possible with 100% efficiency for rgEBOV-VP30_GFP_, rgEBOV-VP30_T2A-GFP_, and rgEBOV-L_T2A-GFP_. However, no virus could be rescued for rgEBOV-NP_T2A-GFP_ and rgEBOV-L_GFP_, despite 15 rescue attempts in 3 independent experiments.

Virus growth in comparison to rgEBOV of the successfully rescued viruses showed slight attenuation of rgEBOV-VP30_T2A-GFP_ and rgEBOV-L_T2A-GFP_, although rgEBOV-L_T2A-GFP_ reached wild type titers at day 6 (Figure 4b). Surprisingly, rgEBOV-VP30_GFP_ was highly attenuated, even though life cycle modeling systems did not indicate any loss in functionality for VP30-GFP. As expected, fluorescence intensity was less pronounced for rgEBOV-L_T2A-GFP_ than for the VP30-tagged viruses (Figure 4c). When comparing the results from the life cycle modeling systems with the results from virus rescue, it became apparent that in all cases tested attenuation of recombinant viruses was more pronounced than seen in the life cycle modeling systems, i.e., those cases where tagging resulted in no attenuation in the life cycle modeling systems, the corresponding GFP-expressing viruses were either attenuated or couldn’t be rescued, and in the case where tagging resulted in attenuation in the life cycle modeling systems, rescue was not possible (Figure 4d).

## 4. Discussion

Life cycle modeling systems are convenient tools to study highly pathogenic viruses such as EBOV under BSL-1 or -2 conditions. In particular, the 4-cis minigenome system often generates data very similar to results of infection experiments. For example, in a study testing an antiviral drug blocking filovirus entry, infection with trVLPs or virus showed very similar IC_50_ values in vitro in the low micromolar range [28,29]. Similarly, when analyzing the role of late domains in VP40, the 4-cis minigenome system yielded results that were much more consistent with infection data than were results obtained with classical virus-like particles that had been produced through overexpression of VP40 [30,31]. Nevertheless, there are also examples where predictions based on life cycle modeling systems could not be confirmed in infection. Such an example is the role of MARV VP30 for the virus life cycle, where both minigenome and trVLP systems indicated that MARV VP30, while involved in viral transcription, was not absolutely essential in the virus life cycle, although for rescue of MARV from cDNA it was absolutely required [32,33,34,35].

In this study, we have shown that life cycle modeling systems are useful for predicting attenuation of viruses harboring smaller tags. However, addition of larger tags, such as GFP, has a more severe impact on infectious viruses than would be expected based on life cycle modelling systems. Importantly, in none of the tested tagging approaches recombinant viruses were less attenuated than expected based on the life cycle modeling system data (and in particular the p1 data from the 4-cis minigenome assay), highlighting the usefulness of these systems in excluding non-viable tagging approaches.

The limitation of life cycle modeling systems to predict the effect of larger tags might be due to the fact that RNP proteins are overexpressed in these systems. Therefore, suboptimal protein–protein interactions that could affect protein function or incorporation of viral proteins into trVLPs, e.g., due to steric hindrance by the added tag, might be compensated by the sheer amount of available RNP proteins in life cycle modeling system. A corresponding virus, however, would not be rescuable, as the tagged RNP protein would only be expressed at native levels. An example for this is tagging of L with mCherry in a flexible linker region [14]. Here, reporter activity in minigenome assays was reduced, but could be restored to levels observed with wild type L by increasing the amount of expression plasmid for L-mCherry during transfection. Besides steric hindrance, additional explanations for the reduction in functionality of the tagged proteins could be misfolding, reduced expression levels, or changes in cellular distribution resulting in localization outside of inclusion bodies, where RNA synthesis takes place, and at least some of these situations could be compensated (and thus masked in life cycle modeling systems) by overexpression.

A number of strategies have been pursued to generate reporter protein-expressing recombinant filoviruses. Most of these have been based on the insertion of a reporter gene as an additional transcriptional unit (ATU) [11,17,36,37], or on direct tagging of a viral protein [14,16]. However, most of these viruses show already in vitro a clear attenuation, and the few recombinant EBOV encoding a reporter protein tested in vivo were all attenuated [17]. An additional approach trying to overcome the functional impairment of directly tagged viral proteins has been to express the tagged protein from an ATU in addition to the non-tagged wild type protein. This has been done in case of fluorescently tagged MARV VP40; however, also here, the recombinant virus was attenuated in vitro [15]. Therefore, the search for a tagging strategy resulting in a reporter-expressing EBOV that is not attenuated in vivo needs to be continued.

In conclusion, we have shown that while life cycle modelling system can be used to predict attenuation of viruses containing small tags, they are limited in their use to predict attenuation due to larger tags such as GFP, although they remain useful in excluding non-viable tagging strategies, and that future studies will be required to find tagging strategies that allow the generation of non-attenuated reporter-expressing EBOV.

## Figures and Tables

**Figure 1 viruses-14-01044-f001:**
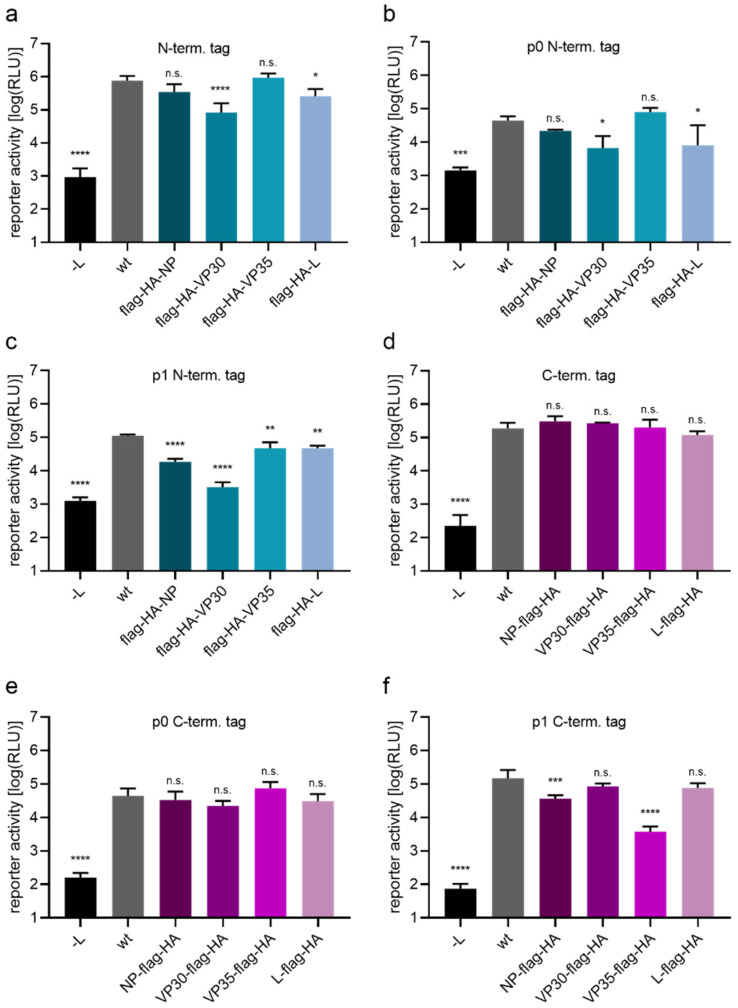
Functionality of flag-HA tagged EBOV proteins. (**a**–**c**) N-terminal tagging of RNP proteins. 293T cells were transfected with all necessary plasmids for a 1- or 4-cis minigenome assay, encoding either untagged RNP proteins (wt) or flag-HA-tagged version as indicated. As a negative control the polymerase L was omitted and replaced by empty vector (-L). Reporter activities in a 1-cis minigenome assay (**a**) or p0 cells (**b**) or p1 cells (**c**) of a 4-cis minigenome assay are shown. (**d**–**f**) C-terminal tagging of RNP proteins. 293T cells were transfected as described in (**a**–**c**). Reporter activity in a 1-cis minigenome assay (**d**) or p0 cells (**e**) or p1 cells (**f**) of a 4-cis minigenome assay are shown. Means and standard deviation of four (**a**,**d**–**f**) or three (**b**,**c**) biological replicates are shown. Significant differences compared to wt are indicated: n.s.: not significant; *: *p* < 0.5; **: *p* < 0.01; ***: *p* < 0.001; ****: *p* < 0.0001.

**Figure 2 viruses-14-01044-f002:**
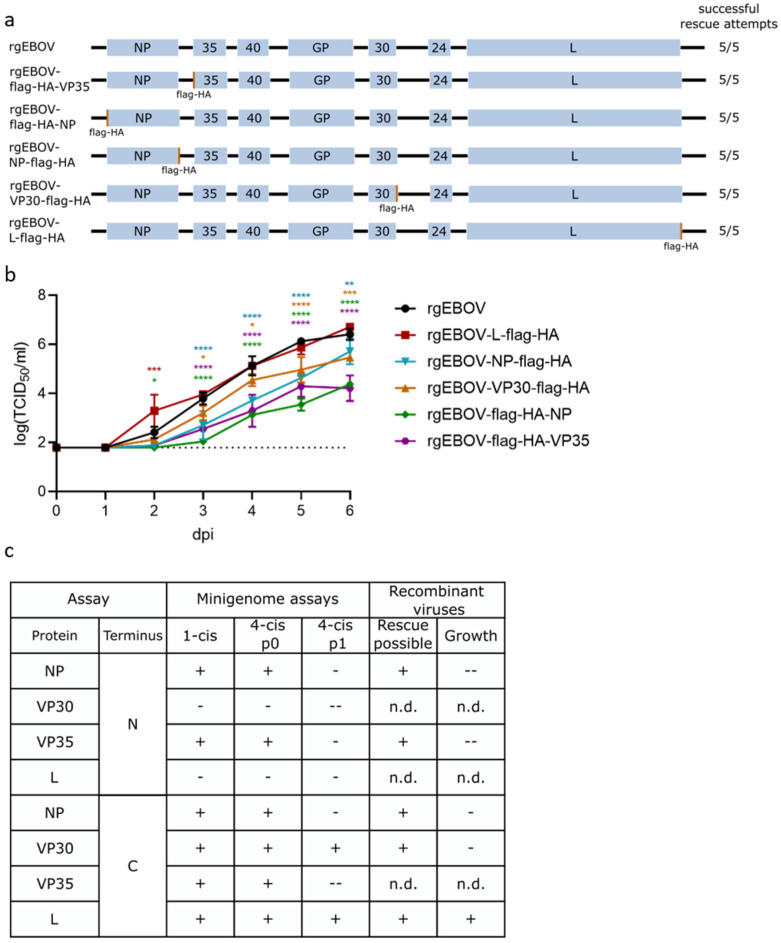
Rescue and characterization of recombinant EBOVs encoding flag-HA-tagged proteins. (**a**) Genome structure of rescued recombinant viruses and rescue efficiency. (**b**) Growth kinetic of rescued viruses. VeroE6 cells were infected with 1 × 10^4^ TCID_50_ of recombinant EBOV with flag-HA-tags at the N- or C-terminus (as noted) of the RNP protein or wild type virus (rgEBOV). Supernatant was harvested at indicated time points and titrated on VeroE6 cells to determine the viral titer. The lower limit of detection is indicated by a dotted line. Means and standard deviations of three biological replicates are shown. Significant differences compared to wt are indicated: n.s.: not significant; *: *p* < 0.5; **: *p* < 0.01; ***: *p* < 0.001; ****: *p* < 0.0001. (**c**) Summary of minigenome (from Figure 1) and virus data for comparison. Symbols indicate whether results were comparable to the positive control or wild type (+; no significant reduction in reporter activity or mean reduction in titers in growth kinetics), slightly reduced (-; significant but less than 1 log reduction in reporter activity or mean reduction in titers in growth kinetics), or highly reduced (--; more than 1 log reduction in reporter activity or mean reduction in titers in growth kinetics). n.d.: not done.

**Figure 3 viruses-14-01044-f003:**
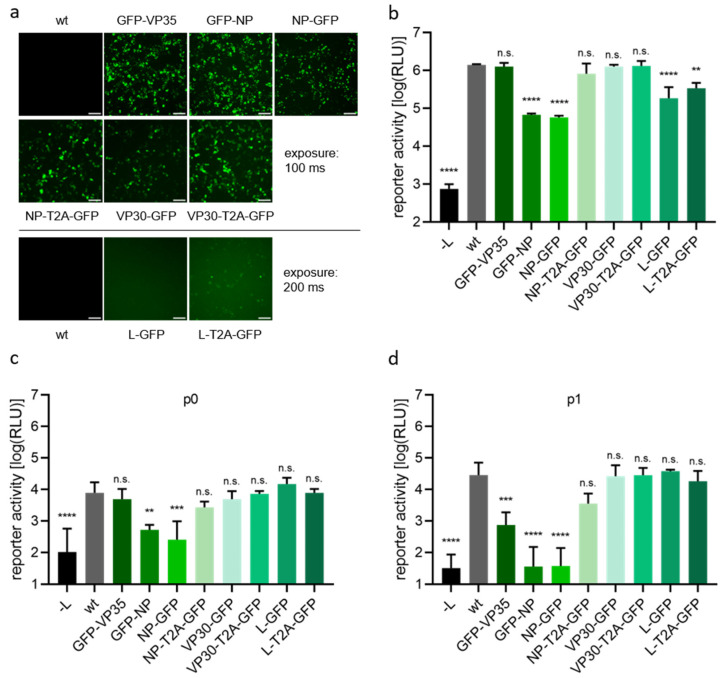
Characterization and functionality of GFP-tagged EBOV RNP proteins. (**a**) Visualization of fluorescence. 293T cells were transfected with all components for a 4-cis minigenome assay. Indicated expression plasmids of RNP proteins were replaced by expression plasmids encoding RNP proteins directly tagged with GFP or proteins GFP-tagged via a T2A site. 48 h after transfection expression of GFP was analyzed by fluorescence microscopy. Scale bar: 100 µm. (**b**) Analysis of protein functionality in a 1-cis minigenome assay. 293T cells were transfected with all components for a 1-cis minigenome assay, including expression plasmids for tagged RNP proteins instead of wild type (wt) proteins as indicated. Cells were lysed 48 h after transfection, and minigenome reporter activity was measured. As a negative control, expression plasmid for L was omitted (-L). (**c**,**d**) Analysis of protein functionality in a 4-cis minigenome assay. 293T cells were transfected with all components for a 4-cis minigenome assay. Lysis and measurement of reporter activity in p0 cells (**c**) was performed 72 h after transfection, while p1 cells were assessed 72 h after infection (**d**). The polymerase expression plasmid was replaced by empty vector as a negative control (-L). Means and standard deviations of three biological replicates (**b**–**d**) or a representative of three biological replicates (**a**) are shown. Significant differences compared to wt are indicated: n.s.: not significant; **: *p* < 0.01; ***: *p* < 0.001; ****: *p* < 0.0001.

**Figure 4 viruses-14-01044-f004:**
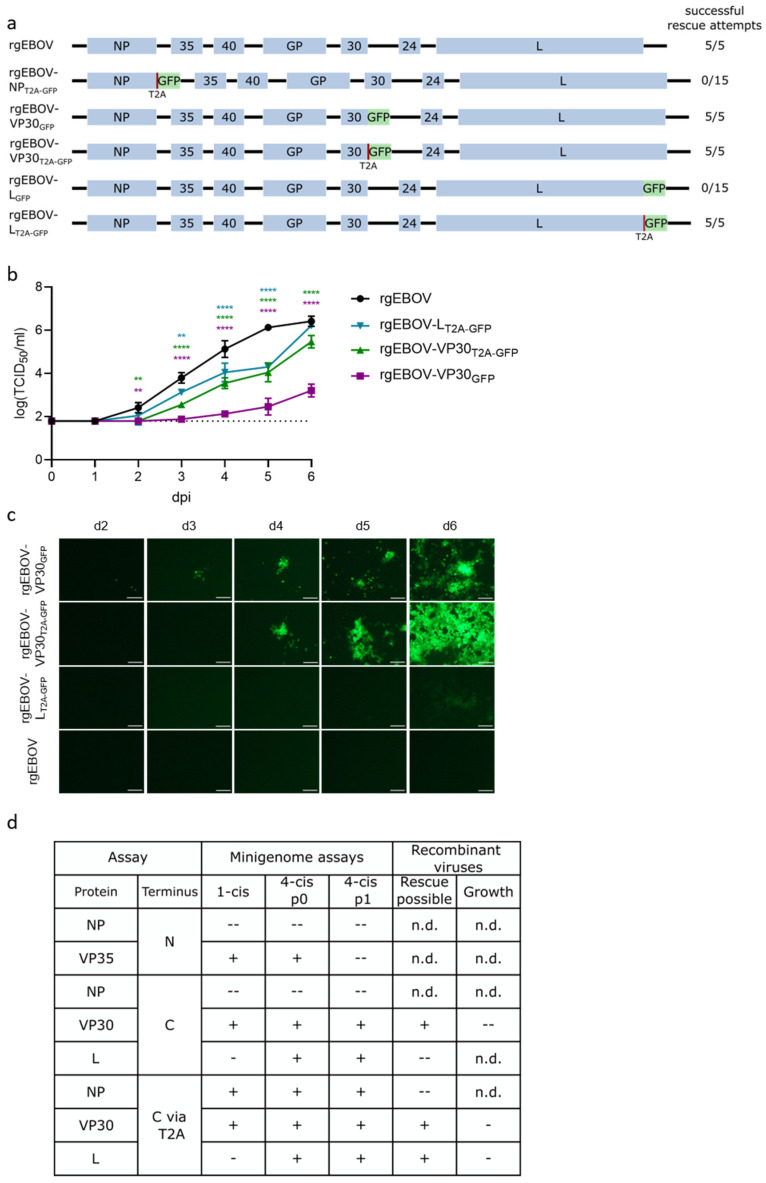
Characterization of GFP- and T2A-GFP-tagged recombinant EBOV. (**a**) Overview of genome structure of recombinant EBOV constructs encoding GFP as a reporter used for rescue attempts. (**b**) Growth kinetics of rescued viruses. 293T cells were infected with 1 × 10^4^ TCID_50_ of either GFP-tagged, GFP via a T2A-site tagged (as depicted), or untagged (rgEBOV) recombinant viruses. The lower limit of detection is indicated by a dotted line. Means and standard deviations of three biological replicates are shown. Significant differences compared to wt are indicated: n.s.: not significant; **: *p* < 0.01; ****: *p* < 0.0001. (**c**) Visualization of reporter activity. GFP-signals in the experiment described in (**b**) were visualized by fluorescence microscopy at depicted time points (scale bar 100 µm). (**d**) Summary of minigenome (from Figure 3) and virus data for comparison. Symbols indicate whether results were comparable to the positive control or wild type (+; no significant reduction in reporter activity or mean reduction in titers in growth kinetics), slightly reduced (-; significant but less than 1 log reduction in reporter activity or mean reduction in titers in growth kinetics), or highly reduced (--; more than 1 log reduction in reporter activity or mean reduction in titers in growth kinetics, or rescue not possible). p: passage; n.d.: not done.

## Data Availability

Not applicable.

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
