# Peer review of "Assessment of Life Cycle Modeling Systems as Prediction Tools for a Possible Attenuation of Recombinant Ebola Viruses"

_viruses, 2022, doi:10.3390/v14051044_

Round 1

Reviewer 1 Report

The work seeks to use the author's minigenome systems as a screening system to weed out fusion strategies that would stand low chances of success in a full-length reverse genetics rescue approach.  The data suggests the authors have been successful in excluding such non-viable tagging approaches.  The study will be of interest to those in the fields of engineering recombinant filoviruses and represents a substantial body of work, especially driving through minigenomes to full-length rescues.  While not particularly innovative it is certainly a very important contribution to the field.

Perhaps beyond the scope of the work is what the consequences of fusion were on the characteristics of the particular protein chosen...did the tags impact expression levels, oligomerization, solubility, cellular distribution....and could these answers yield information on the mechanisms for some of the failures?  A comment on this might be pertinent.

The C-terminal L and VP30 tagging appears to be a potentially promising avenue to pursue from the flag-HA perspective and even the T2A-GFP fusions.  That is very exciting.  I was a little unclear on what the linker was between VP30/L and GFP in the non-2A cleavage system....is it long and flexible to allow independent folding of both proteins?  It would be helpful to clarify the nature of the linker.  These look like interesting sites to perhaps evaluate other monomeric fluorescent/nLuc style reporters to empirically test whether "expressability"...folding/maturation characteristics might improve upon GFP.  Some of the brighter fluorescent proteins might also give options to salvage the L system's low brightness, presumably due to much lower expression levels, in fig 4.c relative to VP30.  

With lines 82 and 86, is it known whether the 2A skipping is always 100 % ? It would be helpful to state if this is known to date from other folks' work. Although I would not ask for this data in the current manuscript, some kind of future SDS-PAGE or MS analysis might be worth looking into just to confirm.  For if there is still some fusion protein that could be more deleterious than the two separate proteins, as envisaged, it would be important to know since the non-T2A versions are compromised.

Minor text question: line 38, why add "respectively"?  It would make sense if you are mentioning different geographies but it seems redundant as written.

The only option that was suitable to tick was: "English language and style are fine/minor spell check required", but to be honest I wasn't aware of any spelling errors.

Reviewer 2 Report

In this paper the authors conduct a study on the assessment of life cycle modeling systems as prediction tools for a possible attenuation of recombinant Ebola viruses

Recombinant EBOVs are important to study antivirals against EBOV, better understand life cylcl and pathogenesis. However, they all show, at least in vivo, attenuation compared to wild type virus, and the basis 13 of this attenuation is only poorly understood. Compared to wildtype viruses, these recombinants are often attenuated and it is important to understand the mecanisms of these attenuantion because studying wildtype EBOV viruses requires BSL4 facilities in contrast to recombinant EBOV which can be studied in BSL1 and 2 facilities.

Overall, the study is well conducted. The authors tested different tag systems and compared to wildtype strains and showed that life cycle modelling system can predict attenuation of viruses containing small tags, but when using larger tags, they their use to predict remains limited. The authors conclude thus that future studies are thus still required to find tagging strategies for generation of non-attenuated reporter-expressing EBOV.

The aim of the study is of major interest and important to allow testing or antivirals to BSL4 pathogens in other BSL conditions. The authors should better precise why chose certain tags and better develop  the practical impact of their observations.

The authors should alos better explain what kind of studies can be conducted with small tags? And which kind of studies cannot be done yet.

Reviewer 3 Report

In this manuscript Bodmer and Hoenen characterize the effect of N- or C-terminal tags on replication and virus rescue.  impact of. Using life cycle modeling systems, they assess the impact of flag-HA or GFP tags on various EBOV proteins. In figure 1, The authors test the functionality of flag-HA tagged proteins using 1-cis and 4-cis minigenome assays. In figure 2, tagged variants with the least impact were identified and rescued. The growth kinetics of the rescued viruses was then assessed. In Figure 3, GFP-tagged variants are assessed using the minigenome systems, while fig 5, characterizes the rescued virus growth kinetics.   

Overall, this was an interesting paper and a well-organized manuscript.  I have no serious concerns with the methodologies or any major issues with the manuscript in general.

However, a minor comment needs to be addressed:

  • Can the authors comment on the overall trends across the two systems? Tagging of NP and VP35 (attenuation with flag-HA while GFP-tags made them non-functional) seems to have much more of an impact on viability than VP30 or L. A few lines in the discussion talking about the possible mechanism of this inhibition will be useful to a non-specialized reader.
